# Cyclic Voltammetry in Biological Samples: A Systematic Review of Methods and Techniques Applicable to Clinical Settings

**Hsiang-Wei Wang** [1], **Cameron Bringans** [1], **Anthony J. R. Hickey** [2], **John A. Windsor** [1], **Paul A. Kilmartin** [3] and **Anthony R. J. Phillips** [1,2,*]

1. Surgical and Translational Research Centre, Department of Surgery, School of Medicine, Faculty of Medical and Health Science, The University of Auckland, Auckland 1142, New Zealand; hwan059@aucklanduni.ac.nz (H.-W.W.); cameronbringans@gmail.com (C.B.); j.windsor@auckland.ac.nz (J.A.W.)
2. School of Biological Science, Faculty of Science, The University of Auckland, Auckland 1142, New Zealand; a.hickey@auckland.ac.nz
3. School of Chemical Science, Faculty of Science, The University of Auckland, Auckland 1142, New Zealand; p.kilmartin@auckland.ac.nz
* Correspondence: a.phillips@auckland.ac.nz

**Abstract:** Oxidative stress plays a pivotal role in the pathogenesis of many diseases, but there is no accurate measurement of oxidative stress or antioxidants that has utility in the clinical setting. Cyclic Voltammetry is an electrochemical technique that has been widely used for analyzing redox status in industrial and research settings. It has also recently been applied to assess the antioxidant status of in vivo biological samples. This systematic review identified 38 studies that used cyclic voltammetry to determine the change in antioxidant status in humans and animals. It focusses on the methods for sample preparation, processing and storage, experimental setup and techniques used to identify the antioxidants responsible for the voltammetric peaks. The aim is to provide key information to those intending to use cyclic voltammetry to measure antioxidants in biological samples in a clinical setting.

**Keywords:** oxidative stress; reactive oxygen species; antioxidant; cyclic voltammetry; biological samples; human; animals





## 1. Introduction

Oxidative stress (OS) plays an important role in the pathogenesis of many diseases and associated complications [1,2], and it has drawn widespread attention as a target for medical interventions [3–5]. OS is the imbalance between the levels of reactive oxygen species (ROS) (such as hydroxyl, superoxide and peroxides) and antioxidants (such as ascorbic and uric acid) present in biological systems [6]. ROS are highly volatile and denature DNA, protein, and lipid [7]. Antioxidants tightly regulate ROS by direct scavenging of ROS, inhibiting ROS production and/or up-regulating other antioxidants [8]. Excess build-up of ROS can overwhelm the antioxidant defences resulting in cell death, tissue injury, organ dysfunction and adverse patient outcomes [8,9].

These observations highlight the need for a practical way to measure and monitor oxidative stress and antioxidants in the clinical setting. This would have the potential to make a significant contribution to the management of many diseases, by enabling the early diagnosis of oxidative stress, measuring antioxidant levels, monitoring changes in oxidative stress over time, guiding antioxidant treatment, and even providing a prognostic marker for both acute and chronic diseases [10–15]. The challenge to measurement has always been that ROSs are transient in nature [16]. Further, direct ROS detection using magnetic spin resonance technology cannot be readily adapted to the clinical setting. Thus,

there remains a need to develop a practical technology for measuring and monitoring ROS and OS in patients.

The most common approach to measuring OS is to measure individual antioxidant levels in the same biological system, which tend to decrease in response to OS. Examples of this approach include those demonstrating a strong correlation between the decreased level of antioxidants and the severity of acute pancreatitis [17,18] and intra-abdominal sepsis [19]. While this approach appears attractive, the reality is that there is a wide range of antioxidant compounds, and each of these requires a unique assay method for their measurement [16,20,21]. Furthermore, there is also a lack of consensus among researchers on how to measure and report antioxidant status across a broad spectrum of biological samples and diseases in a way that can yield meaningful and comparable results [22].

Cyclic voltammetry (CV) is a versatile electrochemical technique used to analyze redox status in a wide range of industrial and research settings. The common applications include evaluation of drug quality in pharmaceuticals [23,24], determination of phenolics and antioxidants in food and wine [25], and label-free detection of biomolecules such as hormones [26,27]. Conventional CV measurements are performed with a three-terminal cell configuration consisting of a working electrode (WE), a counter electrode (CE), and a reference electrode (RE) as shown in Figure 1. The technique measures the current response to a sweeping voltage potential applied to the sample via the WE. CV is highly sensitive to the detection of low-molecular-weight antioxidants (LMWA), which play a vital role in defence against OS [28].

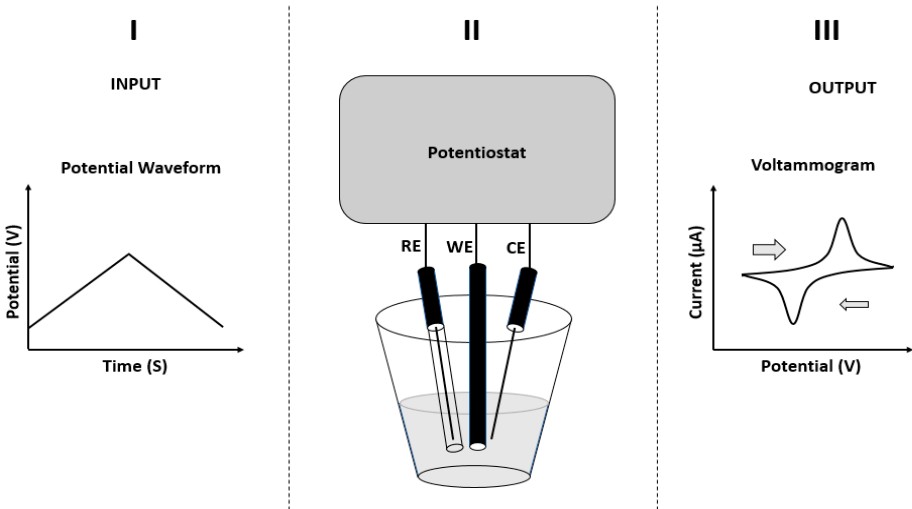

**Figure 1.** (**II**) Conventional cyclic voltammetry setup with potentiostat connecting to three electrodes: working electrode (WE), the counter electrode (CE), and the reference electrode (RE). The electrode tips are submerged under the analyte in a sample vial. (**I**) A sweeping voltage potential is applied by the potentiostat to the analyte via WE; the potential increases and then decreases in the "cycle" at a pre-set duration. (**III**) The current response from analyte is measured at the WE and plotted against applied voltage potential in a voltammogram. The broad arrow indicates anodic current response, and the thin arrow indicates cathodic current response.

Figure 2 shows a typical CV voltammogram of human plasma. The voltammetry peaks are due to the specific redox molecule(s) present in the biological sample, and the amplitude of current at each peak correlates with the concentration of the molecules [29,30]. Traditional electrodes are reusable and can be made of different conductive materials and are formatted in a range of sizes and shapes. Studies have demonstrated CV can be used to assess the change of antioxidant level, disease severity and disease progression in rodent models of chronic and acute diseases, including acute pancreatitis and shock [31–34].

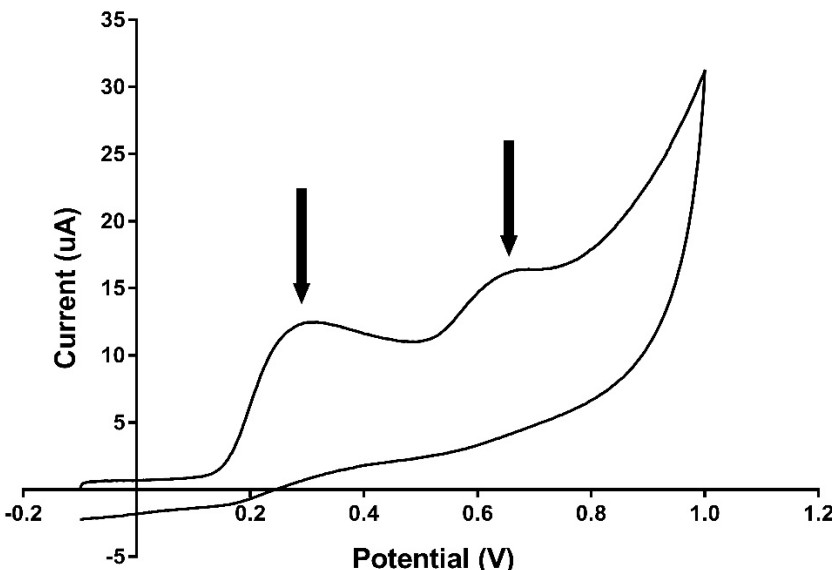

**Figure 2.** Example of cyclic voltammogram of human plasma with glassy carbon WE, Ag/AgCl RE and platinum CE. Scan range: −0.1 V to 1 V, as a scan rate of 100 mV/s. Black arrows indicate two intrinsic voltammetry peaks corresponding to redox-active molecules in the sample (see Section 3.3).

While there have been some reviews on how to use CV to measure biological samples, they have been cursory [28–30,35]. With the significant developments in the field of electrochemistry, this systematic review aims to examine all of the published studies to produce a detailed description and evaluation of CV methods for measuring antioxidant status in biological systems. As such, this paper will review the factors that are fundamental to the use of CV and will identify areas that require further validation and research.

## 2. Materials and Methods

A systematic search of all relevant studies was conducted according to the PRISMA statement [36]. An electronic literature search was performed on SCOPUS, EMBASE, PUBMED, and GOOGLE SCHOLAR for all related studies published before 1 November 2019. The search terms were carefully selected as "Voltammetry" AND "Oxidative Stress" OR "Oxidative Damage" OR "Reactive Oxygen Species" OR "Free Radicals" OR "Antioxidant" OR "Reducing Agent" AND "Disease" OR "Medicine" OR "Surgery" OR "Ischemia" OR "Neoplasia" OR "Infection" OR "Infarction" OR "Diabetes" OR "Injury" OR "Trauma" OR "Inflammation" OR "Tumour" OR "Cancer" to generate a combined result. The only restriction applied to the search was to exclude "Non-English" studies. Once the primary search and selection were completed, a secondary search was conducted on the references section of all the enrolled studies to search for other relevant publications.

### 2.1. Inclusion and Exclusion Criteria

Both of the co-first-authors reviewed all the studies independently. Duplications were excluded using an automated function on EndNote X6[TM]. The titles and abstracts were reviewed to identify all studies that used cyclic voltammetry to assess in vivo changes in antioxidants to reflect oxidative stress induced by "disease, ageing or treatment". Reviews and letters were also excluded to avoid duplication of information. Articles were excluded if: 1. The disease state or treatment was not defined. 2. The oxidative stress was not in vivo. 3. The voltammetry was used for quantitative assessment of a single antioxidant or neurotransmitter rather than as a global assessment of antioxidant status. 4. The voltammetry

assessment was dependent on a redox treatment agent to induce OS rather than allowing physiological change of antioxidant status.

*2.2. Data Extraction*

The data were extracted from the eligible studies into a data summary template developed by the authors. The data include details of the publication, study organism, sample types, buffers and additives, sample processing/storage, types of electrodes, voltammetry parameters and controls, and list of antioxidants that are responsible for the anodic peaks. This current review focuses on the methodologies and cyclic voltammetry on assessing biological samples. The specific clinical and medical applications of voltammetry were beyond this current review and will be the subject of a further publication.

**3. Results**

There were 1658 publications identified by the search strategy, and 38 studies were included in the synthesis of the data summary table. The selection process is outlined in the PRISMA flowchart (Figure 3). Notably, Kohen and colleagues from the Hebrew University (Israel) have published 27 studies in a broad range of disease settings and interventions from 1992 to 2018 [33,37–62]. Two in vitro studies [63,64] and five retrospective studies reviews [28–30,35,65] from this group are excluded from the synthesis of data summary table because they did not meet the inclusion criteria and to avoid duplication of information. Pohanka and colleagues from the Centre of Advanced Studies, University of Defence, (Hradec Králové, Czech Republic) published five studies from 2009 to 2011 [66–70]. This group have also published three studies utilizing Square Wave Voltammetry (SWV) and Differential Pulse Voltammetry (DPV) [71–73]. These techniques have not been included in this review, as they represent significant departures from the standard CV [74]. Mittal and colleagues from the University of Auckland, New Zealand published two studies examining CV in the context of acute pancreatitis and shock [31,32]. Four other research groups published the remaining four studies from Italy, the Czech Republic, France and India, respectively [34,75–77].

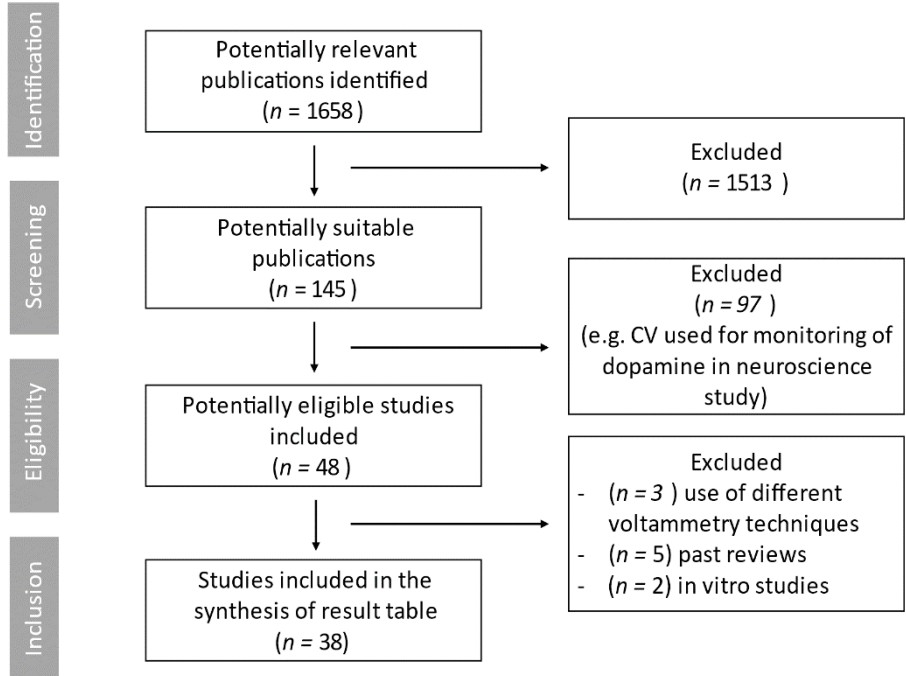

**Figure 3.** PRISMA flowchart for article identification & selection.

### 3.1. Sample Sources, Processing, and Storage

Two-thirds of the studies were animal-based (n = 25), and the remaining 13 studies consisted of 12 human studies, with one combined animal/human study.

#### 3.1.1. Blood Sampling

Blood plasma was primarily used in human studies (n = 8 studies) and less than half of the animal studies (n = 10). A small number of animal-based studies employed serum (n = 2) and whole blood (n = 1). To facilitate sample handling and plasma extraction, two anticoagulants were used: Ethylenediaminetetraacetic (EDTA) (n = 4, human studies) and Heparin (n = 14). Quality control measures for blood sampling, such as sample haemolysis tests, were not reported in any studies. Blood samples have been measured neat, where the electrolytes required for the measurement are provided intrinsically. If the blood samples were neatly measured, the electrolytes needed for the measurement are supplied intrinsically. Alternatively, some researchers have reported using the Phosphate Buffer Saline (PBS) or KCl to dilute their sample [31,32,34,76]. In the case of PBS, it will supply the measurement with additional electrolytes, including NaCl, KCl, $Na_2HPO_4$, $KNPO_4$.

#### 3.1.2. Tissue Samples

Table 1 lists the wide range of solid organs tested using CV. Except in one study where the antioxidant status of skin was assessed by direct skin-electrode contact [75], all other studies reported homogenizing solid tissue in an aqueous buffer solution. PBS was used by 19 studies and a single study reported using Tris-buffer to aid homogenization of brain tissue [67]. One study reported using an ice bath to mitigate the impact of heat produced during the homogenization process [33]. Two studies reported using a centrifuge to remove debris and large blocks of tissue from the homogenate [41,61].

**Table 1.** Summary of studies with published information on organisms, sample type and voltammetry technique details.

| Author (Country) (Ref) | Year | Organism | Sample Type | Sample Storage | Working Electrodes | Counter Electrodes | Reference Electrodes | Scan Range | Scan Rate | Sample Size |
|---|---|---|---|---|---|---|---|---|---|---|
| Lenchner (Israel) [62] | 2018 | Dog | Blood plasma | Stored at −80 °C | Glassy carbon disc | Pt wire | Ag/AgCL | #~1.3 V | 100 mV/s | not specified |
| Devkar (India) [77] | 2016 | Rat | Liver | N/A | Glassy Carbon | Pt Wire | Saturated Calomel | −0.3 V–1.3 V | 400 mV/s | 2 mL of liver/PBS homogenates (20% *w/v*) |
| Pohanka (Czech) [66] | 2011 | Rat | Blood plasma | Fresh sample tested | Graphite * | Graphite | Ag/AgCl | −0.1~1.1 V | 50 mV/s | 20 microliter |
| Mittal (NZ) [32] | 2010 | Rat | Blood serum | Aliquot and stored at −80 °C. | Glassy Carbon | Pt wire | Ag/AgCl | −0.1~1.2 V | 100 mV/s | 4× dilution with PBS to 1 mL |
| Pohanka (Czech) [67] | 2009 | Rat | Blood plasma/brain homogenate | Storage of plasma sample unclear, brain sample stored at −80 °C | Graphite * | Platinum | Ag/AgCl | −0.4~0.9 V | 10 mV/s | 20 microliter |
| Pohanka (Czech) [68] | 2009 | Rat | Blood plasma | not specified | Platinum * | Platinum | Ag/AgCl | −1~1 V | 50 mV/s | 20 microliter |
| Koren (Israel) [37] | 2009 | Human | Blood plasma | Stored at −80 °C | Glassy Carbon | Pt wire | Ag/AgCl | −0.3~1 V | 100 mV/s | Not specified |
| Bandouchova (Czech) [70] | 2009 | Mouse/Vole | Blood plasma | Stored at −20 °C and tested within few days. | Platinum * | Platinum | Ag/AgCl | Not specified | 100 mV/s | 20 microliter |
| Pohanka (Czech) [69] | 2009 | Rat | Blood plasma | no specified | Platinum * | Platinum | Ag/AgCl | −0.5~1.1 V | 50 mV/s | 20 microliter |
| Mittal (NZ) [31] | 2008 | Rat | Blood serum | Stored at −80 °C | Glassy carbon | Pt wire | Ag/AgCl | −0.1~1.2 V | 100 mV/s | 4× dilution with PBS to 1 mL |
| Ruffien-Ciszak (France) [75] | 2008 | Human | Skin | N/A | Gold and Platinum microelectrodes | Pt wire | Saturated Calomel | Gold—0.3~1.5 V Platinum −0.4~1.2 V | 50 mV/s | Direct skin measurement |

**Table 1.** *Cont.*

| Author (Country) (Ref) | Year | Organism | Sample Type | Sample Storage | Working Electrodes | Counter Electrodes | Reference Electrodes | Scan Range | Scan Rate | Sample Size |
|---|---|---|---|---|---|---|---|---|---|---|
| Ryu (Israel) [60] | 2008 | Rat | Brain, lung, liver, heart, pancreas, spleen, kidney homogenate and blood plasma | stored at −80 °C | Glassy Carbon | Pt wire | Ag/AgCl | −0.3~1.3 V | 100 mV/s | Not specified |
| Beit-yannai (Israel) [39] | 2007 | Rabbit | Aqueous Humour | stored at −70 °C | Glassy Carbon | Pt wire | Ag/AgCl | −0.3~1.3 V | 100 mV/s | Not specified |
| Panikashvili (Israel) [59] | 2006 | Rat | Brain homogenate | not specified | Glassy Carbon | Pt wire | Ag/AgCl | 0~1.3 V | 100 mV/s | Not specified |
| Beni (Israel) [40] | 2006 | Rat | Brain cortical, cerebella and liver homogenate | not specified if fresh sample tested or stored. | Glassy Carbon | Pt wire | Ag/AgCl | 0~1.3 V | 100 mV/s | 120 mg of tissue homogenates in PBS 10:1 (W:V) |
| Glantz (Israel) [48] | 2005 | Rat | Brain, heart, liver and lung homogenate | not specified | Not specified | Not specified | Ag/AgCl | 0~1.3 V | 100 mV/s | Not specified |
| Ligumsky (Israel) [56] | 2005 | Human | Gastric juice | stored at −70 °C | Glassy Carbon | Pt wire | Ag/AgCl | not specified | 100 mV/s | 1 mL to 1 mL dilution with PBS |
| Beni (Israel) [41] | 2004 | Mouse | Brain cortex homogenate | not Specified | Glassy Carbon | Pt wire | Ag/AgCL | 0~1.3 V | 100 mV/s | Not specified |
| Granot (Israel) [50] | 2004 | Human | Blood plasma (EDTA) | stored at −70 °C | Glassy Carbon | Pt wire | Ag/AgCl | −0.3~1.3 V | 100 mV/s | Not specified |
| Kohen (Israel) [55] | 2004 | Human/Rat | Skin secretion | Not specified, if only fresh sample tested | Glassy Carbon | Pt wire | Ag/Agcl | 0~1.3 V | 100 mV/s | 0.5 mL |
| Granot (Israel) [49] | 2002 | Human | Blood plasma (EDTA) | stored at −70 °C | Glassy Carbon | Pt wire | Ag/AgCl | −0.3~1.3 V | 100 mV/s | Not specified |

**Table 1.** *Cont.*

| Author (Country) (Ref) | Year | Organism | Sample Type | Sample Storage | Working Electrodes | Counter Electrodes | Reference Electrodes | Scan Range | Scan Rate | Sample Size |
|---|---|---|---|---|---|---|---|---|---|---|
| Mantovani (Italy) [34] | 2002 | Human | Blood plasma | stored at −20 °C | Double Pt wire | Double Pt wire | Ag/AgCl | −0.3~1.3 V | 100 mV/s | 7 mL sample + 1 mL of 0.8 M KCL |
| Nitzan (Israel) [58] | 2002 | Human | Synovial fluid/saline aspirate | Stored at −20 °C | Glassy Carbon | Pt wire | Ag/AgCl | 0~1.3 V | 100 mV/s | Not specified |
| Psotova (Cezch) [76] | 2001 | Human | Blood plasma (EDTA) | not specified | Glassy Carbon | Pt wire | Calomel saturated electrode | −0.4~0.8 V | 200 mV/s | Plasma 0.3 mL + PBS 1.5 mL |
| Green (Israel) [52] | 2001 | Rat | Brain homogenate | not specified | Glassy Carbon | Pt wire | Ag/AgCl | 0~1.3 V | 100 mV/s | 0.5 mL |
| Blau (Israel) [42] | 2000 | Rat | Colonic mucosal homogenate | stored at −70 °C | Glassy Carbon | Pt wire | Ag/AgCl | 0~1.3 V | 100 mV/s | Not specified |
| Dubnov (Israel) [45] | 2000 | Mouse | Brain, heart, lung, spleen, liver, small bowel, kidney, quadriceps muscle homogenates and blood plasma | stored at −70 °C | Glassy Carbon | Pt wire | Ag/AgCl | 0~1.3 V | 100 mV/s | Not specified |
| Elangovan (Israel) [47] | 2000 | Rat | Brain homogenate | stored at −70 °C | Glassy Carbon | Pt wire | Ag/AgCl | 0~1.3 V | 100 mV/s | Not specified |
| Elangovan (Israel) [46] | 2000 | Rat | Brain, liver, heart, kidney homogenate and blood plasma | stored at −70 °C | Glassy Carbon | Pt wire | Ag/AgCl | 0~1.3 V | 100 mV/s | Not specified |
| Shohami (Israel) [61] | 1999 | Rat | Brain, heart, liver, lung, kidney, intestine and skin homogenate | Snap frozen using liquid nitrogen then stored at −70 °C before final processing | Glassy Carbon | Pt wire | Ag/AgCl | 0~1.3 V | 100 mV/s | Not specified |

**Table 1.** *Cont.*

| Author (Country) (Ref) | Year | Organism | Sample Type | Sample Storage | Working Electrodes | Counter Electrodes | Reference Electrodes | Scan Range | Scan Rate | Sample Size |
|---|---|---|---|---|---|---|---|---|---|---|
| Chevion (Israel/Germany) [44] | 1999 | Human | Blood plasma | stored at −80 °C | Glassy carbon | Pt wire | Ag/AgCl | −0.3~1.3 V | 100 mV/s | Not specified |
| Granot (Israel) [51] | 1999 | Human | Blood plasma (EDTA) | Stored at −70 °C in nitrogen gas | Glassy carbon | Pt wire | Ag/AgCl | −0.3~1.3 V | 100 mV/s | Not specified |
| Beit-Yannai (Israel) [38] | 1997 | Rat | Brain and heart homogenate | not Specified | Glassy Carbon | Pt wire | Ag/AgCl | 0~1.3 V | 100 mV/s | 250 microliters |
| Chevion (Israel) [43] | 1997 | Human | Blood plasma | Stored at −80 °C | Glassy Carbon | Pt wire | Ag/AgCl | −0.3~1.3 V | 100 mV/s | low volume cell |
| Kohen (Israel) [33] | 1997 | Rat | Brain, lung, liver, heart, kidneys and skin homogenate | Organ homogenate stored at −20 °C, skin snap froze by liquid nitrogen then stored at −70 °C | Glassy Carbon | Pt wire | Ag/AgCl | −0.2~1.3 V | Not specified | Not specified |
| Lomnitski (Israel) [57] | 1997 | Mouse | Brain homogenate | −70 °C | Glassy Carbon | Pt wire | Ag/AgCl | 0~2 V | Not Specified | 250 microliters |
| Kohen (Israel) [54] | 1992 | Human | saliva | Not specified | Glassy Carbon | Not specified | Ag/AgCl | 0~2 V or −1.5~2 V | 100 mV/s | PBS dilution to 250 microliters |
| Kohen (Israel) [53] | 1992 | Rat | Skin, Brain, intestinal epithelium, kidney, liver, lung homogenate and whole blood | Stored at −20 °C | Glassy Carbon | Pt wire | Ag/AgCl | 0~2.0 V | Not specified | 1:1 sample: PBS dilution to 250 microliters |

* Screen-Printed Electrode; # No lower range; Ag/AgCl: Silver/Sliver chloride; EDTA: Plasma processed using Ethylenediaminetetraacetic Acid; PBS: Phosphate Buffer Solution; Pt wire: Platinum wire.

### 3.1.3. Lipophilic Low Molecular Weight Antioxidant Extraction

Four studies [45,46,50,61] reported extracting lipophilic LMWA using a modified method described by Motchnik et al. [78]. The extraction solutions used for the process included methanol/hexane mixture, acetonitrile and tetrabutylammonium perchlorate, which were reported to be CV neutral. Granot et al. reported that this method of extraction does not apply to plasma as it causes inconsistent results because lipid-soluble antioxidants are transported by blood lipoprotein [49].

### 3.1.4. Other Sample Types

Antioxidant status from other body fluids including saliva [54], gastric juice [56], synovial fluid [58], skin secretions [55], cerebrospinal fluid (CSF) [53] and aqueous humour from the rabbit eye [39] has been measured. A range of specialized methods and collection protocols were used, including fine needle aspiration (synovial fluid and aqueous humour), gastroscopy (gastric juice) and collection wells (for skin secretion) are used. Two fresh samples of body fluid were "bubbled with nitrogen to eliminate unwanted oxygen" [54,55]. There is no reported application of this technique to blood sample or homogenized tissue.

### 3.1.5. Sample Storage

19 studies reported sample storage at −70 to −80 °C while four studies reported storage at −20 °C and one study recorded fresh specimen testing [67]. In one study, nitrogen gas was used to assist sample stability in storage [51]. It has been documented that tissue sample can be stored at −80 °C for up to 6 months without any apparent impact on the CV [30,33,35].

### *3.2. Voltammetry Equipment, Variables and Setting Parameters*

### 3.2.1. Electrodes and Required Sample Volume

The most widely recorded CV electrode configuration is the combination of a glassy carbon WE, platinum CE and a silver/silver chloride (Ag/AgCl) RE (n = 26). Ruffin-Ciszak used specially fabricated gold and platinum micro-WE to minimize resistivity and increase sensitivity [75]. Pohanka et al. reported on the use of Screen-Printed Electrodes (SPEs) [66,68–71]. Except for three studies that used Saturated Calomel Electrode (SCE) as RE, the remainders opted for Ag/AgCl RE [75–77]. A significant cutback in the required sample volume was noted between standard electrode configuration (0.25 mL to 7 mL) and SPE (20 μL). While sample dilution was not commonly described, Mittal at al. noted that the heights of voltammetry peaks were directly proportional to serial blood serum dilution [31,32].

### 3.2.2. Scan Range

This described the voltage window probed during the measurement. It has been observed that in their earlier studies, Kohen's group set the scan range between −1.5 V~2 V [53,54]. The range was gradually reduced for all studies after 1997 with the upper potential range limited to 1.3 V [33,43]. Similarly, Ruffien-Ciszak et al. used the scan range −0.3~1.5 V for gold microelectrode and −0.4~1.2 V for platinum microelectrode. The group suggested this was to avoid overoxidation of $H_2O$ and subsequent change of pH on the skin surface [75].

### 3.2.3. Scan Rate

Scanning rate determines the rate of voltage ramping, 28 studies reported setting the scan rate to 100 mV/s and four studies setting it to 50 mV/s. One study each used scan rates at 400 mV/s, 200 mV/s and 10 mV/s.

### 3.2.4. Temperature and pH Control

Control of pH was reported by Mittal et al. [31,32] and Ruffin-Ciszak et al. [75]. Control of temperature at 37 °C was reported by Mittal et al. [31,32] and a single study from the Kohen group [53].

### 3.3. Determine the Constituents of Peak Potential

Table 2 lists different LMWAs and methods that have been used to determine the constituents of voltammetric peaks. These results are obtained based on "spiking" the sample with known antioxidants standards, correlation with High-Performance Liquid Chromatography (HPLC) analysis, and enzymatic removal of known antioxidants from the sample.

The first voltammetric peak of water-soluble samples is due to the combination of ascorbic acid and uric acid. This was confirmed by removing the first peak from the sample with the addition of ascorbic oxidase and uricase [55,61]. There is less compelling evidence for the antioxidants responsible for the second peak, with possible candidates being carnosine, cysteine, NADH, NADD(P)H and melatonin [38,61,67].

**Table 2.** Suggested antioxidant composition anodic peaks and method of validation.

| Antioxidants | Reference | Methods of Detection | In Vitro Potential Reported | The Potential Range Where Antioxidant Is Likely to Contribute | Order of Peak on Voltammogram Where Antioxidant Likely to Contribute |
|---|---|---|---|---|---|
| ASCORBIC ACID (H) | Pohanka et al., 2009 [67] | Plasma Spiking | | | |
| | Bandouchova et al., 2009 [70] | Plasma spiking | | 658 mV | 1st |
| | Mittal el al. 2008 [31] | Correlation with standard curve and calculation of theoretical contribution | 450 mV | 450 mV | 1st |
| | Ruffien-Ciszak et al., 2008 [75] | Sample measurement post AA exposure, in vitro measurement | 200–600 mV | 200–600 mV | N/A |
| | Beit-Yannai et al., 2007 [39] | HPLC | | 268.5 +/− 16.29 mV | 1st |
| | Glantz et al., 2005 [48] | correlation with HPLC measurement | | 350 +/− 50 mV | 1st |
| | Kohen et al., 2004 [55] | HPLC, Ascorbate oxidase to reduce the current height | | 476 +/− 49 mV | 1st |
| | Shohami et al., 1999 [61] | Spiking, HPLC, Ascorbate oxidase cause 85% peak reduction. | 330 mV | 320 mV–370 mV | 1st |
| | Beit-Yannai et al., 1997 [38] | Spiking | | 350 +/− 50 mV | 1st |
| ALPHA-TOCOPHEROL (L) | Shohami et al., 1999 [61] | Spiking, HPLC | 855 mV | 932 +/− 107 mV | 2nd |
| BETA-CAROTENE (L) | Shohami et al., 1999 [61] | Spiking, HPLC | 340 mV | 240 +/− 43 mV | 1st |
| CARNOSINE (H) | Shohami et al., 1999 [61] | Spiking | 895 mV | 900 +/− 70 mV | 2nd |
| | Beit-Yannai et al., 1997 [38] | Spiking | | 750 mV | 2nd |
| CYSTEINE | Pohanka 2009 [67] | Spiking | | | 2nd |
| GLUTATHIONE | Pohanka 2009 [67] | Plasma spiking, Used for molar equilibration. | | | |
| | Ruffien-Ciszak et al., 2008 [75] | Measurement post glutathione exposure, in vitro measurement. | 1200 mV | 1200 mV | n/a |
| LIPOIC ACID (L) | Shohami et al., 1999 [61] | Spiking, HPLC | 1080 mV | 932 +/− 107 mV | 3rd |
| MELATONIN (H) (L) | Shohami et al., 1999 [61] | Spiking, HPLC | 755 mV; 870 mV | 900 +/− 70 mV; 932 +/− 107 mV | 2nd |
| | Beit-Yannai et al., 1997 [38] | Spiking | | 750 mV | 2nd |

**Table 2.** *Cont.*

| Antioxidants | Reference | Methods of Detection | In Vitro Potential Reported | The Potential Range Where Antioxidant Is Likely to Contribute | Order of Peak on Voltammogram Where Antioxidant Likely to Contribute |
|---|---|---|---|---|---|
| NADH (H) | Shohami et al. [61] | LDH + pyruvic acid leads 2nd peak reduction | 730 mV | 900 +/− 70 mV | 2nd |
| NADPH (H) | Shohami et al. [61] | Spiking | 720 mV | 900 +/− 70 mV | 2nd |
| TRYPTOPHAN (L) | Shohami et al. [61] | Spiking | 870 mV | 900 +/− 70 mV | 2nd |
| | Beit-Yannai et al., 1997 [38] | Spiking. | | 750 mV | 2nd |
| UBIQUINOL-10 (L) | Shohami et al. [61] | Spiking, HPLC-ECD, | | 240 +/− 43 mV | 1st |
| URIC ACID (H) | Mittal et al., 2010 [32] | Correlated with standard UA concentration curve. | | 350 mV | 1st |
| | Chevion et al., 1997 [43] | Uricase to reduce peak of the sample and comparison with UA spiking standard curve. | 420 mV | 420 mV | 1st |
| | Mittal et al., 2008 [31] | Correlated with standard curve and calculation of theoretical maximum. | 450 mV | 450 mV | 1st |
| | Ruffien-Ciszak et al., 2008 [75] | Sample measurement post UA exposure, in vitro measurement. | 800–1000 mV | 800–1000 mV | n/a |
| | Beit-Yannai et al., 2007 [39] | HPLC | | 268.5 +/− 16.29 mV | 1st |
| | Glantz et al., 2005 [48] | correlation with HPLC measurement | | 350 +/− 50 mV | 1st |
| | Kohen et al., 2004 [55] | HPLC-ECD, Uricase removal of 1st wave | | 476 +/− 49 mV | 1st |
| | Shohami et al. [61] | HPLC-ECD, Uricase cause of 1st peak reduction, Homogenate spiking | 360 mV | 320–370 mV | 1st |
| | Beit-Yannai et al., 1997 [38] | Spiking | | 350 +/− 50 mV | 1st |

## 4. Discussion

This review summarises the published literature on the use of CV for the measurement of antioxidant status in biological samples. The aim is to provide practical information to help clinicians or researchers faced with implementing CV in their research. The technical aspects of CV were systematically examined with particular emphasis on sample collection and handling, experimental and parameter setup, and the various methods used to determine the antioxidant and voltammetric peak connexion. The findings demonstrate the impressive flexibility of CV in being able to analyze a wide range of biological specimens, including blood, homogenized tissue, body fluids and skin (via direct skin-electrode contact). Despite the relative simplicity of the procedure, the use of CV in measuring antioxidant status remains limited and restricted to just a few highly focussed research groups.

Collection of blood through venepuncture is one of the most common and safe clinical procedures, and blood plasma has been used in most of the human studies included in this review. However, poor phlebotomy practises can lead to in vitro haemolysis affecting the quality of the sample and incorrect interpretation of the result [79,80]. Erythrocytes are known to carry both non-enzymatic and enzymatic antioxidants [81], and haemoglobin can interfere with CV analysis [82]. Thus, in vitro, haemolysis may have an impact on the interpretation of CV. It was therefore a surprise, that no group reported quality assurance measures to detect haemolysis. It is recommended that haemolysis tests are done because of the potential confounding by the contamination by intracellular antioxidants and haemoglobin.

While both EDTA and Heparin have been used for anticoagulation of blood sample, EDTA has been reported to interfere with CV readings by oxidizing at a potential of over 900 mV and masking the intrinsic voltammetric peaks. On the contrary, Heparin is not CV sensitive and can be used reliably [29]. Additionally, Heparin has been shown to provide superior preservation capability for ascorbic acid (a key constituent of the first voltammetric peak) over EDTA plasma, serum, citric acid and Stabilyte ® [83]. As a result, Heparin is the ideal anticoagulant for the treatment of blood samples for the measurement of CV.

In addition to compensating for the volume needed for experiments, and preventing coagulation [84], sample dilution can be used to modify the CV sensitivity to identify additional redox-active molecules by altering the surface interaction of the electrodes with the test solution. A study conducted by Kilmartin et al. reported that dilution of red/white wine improves the sensitivity of CV to phenolic groups by enhancing the diffusion of the redox-active molecule to the electrode surface [25]. While the voltammetric peak height of the serum (rat) reported by Mittal et al., is proportional to the sample dilution [31,32], the effect of serial dilution of plasma was not assessed.

Both tissue and plasma samples (at −75 °C) can be stored for at least six months without impact on the CV results [29], with the degradation of frozen brain samples and poor voltammetry signals after that time [28]. Antioxidant stability during storage is an important aspect which has not been adequately addressed in the design of the studies covered by this review. For example, if the plasma sample is insufficiently acidified or if an inadequate temperature is used for long-term storage, plasma ascorbic acid can degrade rapidly [83].

Most of the studies covered in the review used the three standard electrode configurations. Two research groups departed from this convention and introduced microelectrodes and SPEs. Key attributes of the microelectrode are smaller currents, steady-state response, and short response time. This was reported with the measurement of skin antioxidant status by direct skin-electrodes contact. The immunity of microelectrodes to 'ohmic drop' phenomena make it possible to conduct electrochemical experiments in previously inaccessible samples such as non-polar solvents, supercritical fluids and solids [85]. With reference to SPEs, the sample volume required for SPEs is significantly smaller than for standard CV. Another important advantage of SPEs is that they are intended for single-use applications,

eliminating the arduous de-fouling process between each measurement. This also makes them ideal for clinical applications, as SPEs reduce the risk of cross-contamination between the hazardous biological samples and operators [86].

RE is crucial to the accuracy of electrochemical measurements by providing a known stable potential that can be referenced by WE. The ideal RE, such as the Standard Hydrogen Electrode (SHE) has a stable potential set at 0, reversible and non-polarisable with high exchange current density to minimize the risk of noise superimposing the measurement signals [87]. However, as SHE is cumbersome to use and maintain, most researchers chose to use SCE and Ag/AgCl electrodes instead, as we have shown in this study. While SCE has a stable potential at +0.244 V (against SHE) and is simple to construct and maintain, due to the presence of hazardous elemental mercury and Hg2Cl2 (calomel), it has fallen out of fashion [88]. The Ag/AgCl consists of a silver wire dipped in a chloride solution with a standard potential of 0.230 V ($\pm$10 mV). Ag/AgCl electrode non-toxic and easy to use; however, it has some disadvantages, including chlorine ions, that may leak out of the junction and interfere with certain electrode surfaces, such as gold and incompatibility with organic solvents [88]. Such deficiencies have led to an increase in the demand for pseudo-reference electrodes (pseudo-RE) in recent years. These electrodes are mainly structurally solid and made from materials such as elemental silver, platinum or gold wires. They are generally used either in miniaturized electrochemical cells (i.e., portable biosensors) or when organic solvents are used as the measurement media [87]. Some of the advantages of pseudo-RE, including simplicity, no liquid gap junction potential, low ohm resistance, no standard RE solvent contamination. However, the inherent potential of pseudo-RE cannot be measured due to the lack of thermodynamic equilibrium, potential shifts occur because many of them are not ideally nonpolarisable and have limited operating pH or temperature range [89]. Overall, different RE varieties and their respected characteristics could have an impact on and shift the potential of the sample. The RE selection should, therefore, take into account the working electrode, the oxidation potential of the sample and the nature of the measurement environment.

Scanning range and scan rate are two of the most critical parameter setups for CV. It is ideal to scan over a wide range of voltages so that more antioxidants can be measured in the sample. However, a balanced approach must be taken to avoid overoxidation of the electrode surface in aqueous samples at a high potential range. As shown in the result section, most studies included in this review set the scanning range between 0~1.3 V. The exact value will have to be adjusted according to the type of WE and RE used and the antioxidants of interest.

The scan rate is one of the key determinants of current amplitude. In a voltammetry cell, there are two types of electric current: (1) Faradic current (generated when redox-active molecules are oxidized or reduced) and (2) Non-Faradic current (charge between electrode and analyte). Non-Faradic current can be regarded as background noise that limits the sensitivity of the voltammetry system because it can mask the Faradic current. According to the Randles-Sevcik equation [90,91], Faradic current is proportional to the square root of scan rate, whereas the non-Faradic current is directly proportional to the scan rate [92]. Therefore, the scan rate should be selected to provide the optimal balance between Faradic and non-Faradic current. Measuring CV at low scan rates (e.g., 5 mV/s, 10 mV/s) has the advantage of a low non-Faradic current, but a longer scan exposes the electrode to excessive fouling.

In contrast, high scan rates (e.g., 500 mV/s) provide high current with low resistivity in a shorter time frame. However, the large surface of the conventional electrode (2–3 mm) can cause signal distortion at the high scan rate, and so this approach is only suitable for use with microelectrodes [85]. The most commonly used scan rate in this review was 100 mV/s, but there is no evidence provided to support this. There is contrary evidence indicating that the CV sensitivity to ascorbic acid improves with higher scan rate [30]. Individual samples may, therefore, need to be tested with different scan rates, as a higher scan rate might be required to assess samples with a minute concentration of antioxidants

to improve sensitivity. Additionally, by analyzing the sample using different scan rates, Lee et al. were able to selectively detect and estimate homocysteine and glutathione in the presence of cystine by taking advantage of the difference in reaction rates between homocysteine, glutathione and catechol, an essential step in improving the selectivity of CV in a complex biological specimen [93]. This issue remains largely unexplored in biological samples.

Temperature and pH are inter-related and should be controlled close to physiological values for the measurement of biological samples [94,95]. Given the limited evidence available from the current literature, it is strongly recommended that these variables should be measured and controlled. Overall, precise control of all possible parameters, electrode design and processing of samples are required to improve the consistency and sensitivity of the technique. We believe that the field would benefit greatly from standardized experiment design and reporting conventions to make replication of the experiment easier and allow meaningful comparison between studies.

Antioxidants with similar oxidation potential frequently overlap with each other when measured using electrochemical techniques [84]. While this overlap allows an aggregate summary of the LMWA activity of the sample, current CV methods lack the specificity to discriminate individual antioxidants and lack sensitivity to enzymatic antioxidants. This review has shown that HPLC, enzyme, and spiking techniques were used as complementary methods to establish the potential peak-antioxidant connection. Although ascorbic acid and uric acids form the first peak, more work is needed to identify the redox molecules responsible for the second peak [29]. While no studies have reported a method to improve resolution, this limitation is likely to be addressed with the advent of electrode surface modification, altered electron transfer kinetics, and larger electrode surface area [96,97].

This review did not cover studies investigating DPV and SWV. In brief, DPV can improve selectivity when measuring different redox processes when compared with the standard CV. However, DPV also creates practical problems with respect to material stability and cell operation at any larger scale [98]. SWV is the most advanced electrochemical technique, with its advantages including increased sensitivity, short analytical time and the ability to reject non-Faradic current [99]. However, due to its relative complexity, voltammetric data from SWV are less easy to interpret than standard CV, where in most cases, only the net voltammetric peak is analyzed [100].

Consequently, given that CV is a powerful technique for investigating kinetics and mechanisms of the electrochemical process, it is limited because of the lower sensitivity of analyte detection when compared to SWV [101]. An ideal analytical study should therefore begin by performing a CV scan within a large potential window to explore the nature of redox species and determine the potential for oxidation and reduction. This is followed by a highly sensitive SWV scan to analyze the compounds of interest in a smaller, predetermined potential ranges.

Overall, in this current review, given the multidisciplinary nature of CV research, we focused on the technical aspect of using CV with biological samples. The study will likely assist researchers with a clinical background who have difficulty grasping the concept of electrochemistry and those with electrochemistry background who are not experienced in the handling of biological samples.

The clinical evidence of the use of this technique for in vivo oxidative stress or antioxidant status monitoring is not covered in the current review. CV studies have been used for several medical conditions. These include abetalipoproteinemia [50], ageing [33,54,55], anchored temporomandibular joints [58], cancer [34], cerebral ischemia [48], colitis [42], CuZn-SOD deficiency [40], diabetes [46,60], dietary restriction [45], drug toxicity [66–70,77], duodenal ulcer [56], gastric dilatation [62], glycogen storage [62]. Since the majority of studies are limited in the scale, and the research conditions vary considerably, further research is needed to support the translation of the CV technique into full clinical use.

## 5. Conclusions

The technical aspects of the measurement of antioxidant status using CV in biological samples have been reviewed. Practical information about the sample has been provided, including sample types, collection methods, processing and storage. Additionally, provided is information regarding electrode setup, controls and peak composition from various samples, and this provides the basis for several recommendations. There is scope to continue to improve on the performance of CV, which will become a more important tool to the biomedical and electrochemical research community, and it will find many biological and clinical applications.

**Author Contributions:** H.-W.W. and C.B. contributed equally and substantively to this review and shared the first authorship. A.J.R.H., J.A.W., P.A.K. and A.R.J.P. provided supervision, review and editing of the final manuscript. All authors have read and agreed to the published version of the manuscript.

**Funding:** This research was funded by Health Research Council (NZ), grant number 3708109.

**Institutional Review Board Statement:** The study was conducted according to the guidelines of the Declaration of Helsinki, and approved by the University of Auckland Human Participants Ethics Committee on 17 March 2017—Human Ethics Protocol 018755.

**Informed Consent Statement:** Informed consent was obtained from all subjects involved in the study.

**Data Availability Statement:** No new data were created or analyzed in this study. Data sharing is not applicable to this article

**Conflicts of Interest:** The authors declare no conflict of interest.

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
