# Peer review of "Cyclic Voltammetry in Biological Samples: A Systematic Review of Methods and Techniques Applicable to Clinical Settings"

_signals_

Round 1
Reviewer 1 Report
The authors present a review paper on the use of cyclic voltammetry in biological samples. It is clear, concise and informative. As an electrochemist who has not yet worked with biological samples, I learned a lot. I was particularly impressed to see the authors describing exactly how they performed the literature review to shortlist the papers that they would discuss – something quite unusual to see, but helpful.
I think this will be very useful to researchers intending to work in this field. Overall, the review paper is of very high quality and almost suitable for publication. There are a couple of points that could improve the accuracy of the discussion.
- In section 3.2.2, the scan range is discussed, but no mention of the reference electrode material is made. Potential ranges can vary not only when the working electrode is changed, but also the reference electrode. In fact, a stable reference for use in biological media may be difficult to achieve. A discussion on the reference potentials would therefore be useful.
- On page 5, discussing the Faradaic current and electric double-layer current. I suggest that the authors change this to Faradaic and non-Faradaic currents. Non-Faradaic currents can indeed come from double layer capacitance, as they say, but they can also be a result of adsorption/desorption processes.
- Also in this section, it is mentioned that the scan rate should be selected to provide the optimum balance between Faradaic and EDL currents. The first thought I had is that square-wave voltammetry (SWV) could overcome this – i.e. reduce the effect of non-Faradaic currents. I understand that the review is focussed on cyclic voltammetry, but the use of SWV seems obvious for sensitive detection of analytes, and could be a recommendation. This point also relates to the last sentence in the discussion that says that voltammetric data from SWV are less easy to interpret than standard CV. I actually disagree with this. CV is used to study mechanisms and to extract kinetic and thermodynamic parameters, and is very powerful because of this. SWV is used for electroanalysis of low concentrations once you know what peak you are analysing. Of course, the choice of CV vs SWV will depend on how sensitive the measurements are. Some more discussion on this would be beneficial.
- Please also use the conventional abbreviation SWV for square wave voltammetry, not SQV.
- Please remove section 0 and the last part of the discussion section, that are instructions from the journal.
Author Response
Thanks for the time and kind comments you made. As per your advice, we have revised the paper.
- In section 3.2.2, the scan range is discussed, but no mention of the reference electrode material is made. Potential ranges can vary not only when the working electrode is changed, but also the reference electrode. In fact, a stable reference for use in biological media may be difficult to achieve. A discussion on the reference potentials would therefore be useful.
Except for three studies that used Saturated Calomel Electrode (SCE) as RE, the remainders opted for Ag/AgCl RE [75-77]
This was added to line 197-198 by adding more details of RE used by the studies.
RE is crucial to the accuracy of electrochemical measurements by providing a known stable potential that can be referenced by WE. The ideal RE, such as the Standard Hydrogen Electrode (SHE) has a stable potential set at 0, reversible and non-polarisable with high exchange current density to minimise the risk of noise superimposing the measurement signals [87]. However, as SHE is cumbersome to use and maintain, most researchers chose to use SCE and Ag/AgCl electrodes instead, as we have shown in this study. While SCE has a stable potential at +0.244 V (against SHE) and is simple to construct and maintain, due to the presence of hazardous elemental mercury and Hg2Cl2 (calomel), it has fallen out of fashion [88]. The Ag/AgCl consists of a silver wire dipped in a chloride solution with a standard potential of 0.230 V (± 10 mV). Ag/AgCl electrode non-toxic and easy to use, however, It has some disadvantages, including chlorine ions, that may leak out of the junction and interfere with certain electrode surfaces, such as gold and incompatibility with organic solvents [88]. Such deficiencies have led to an increase in the demand for pseudo-reference electrodes (pseudo-RE) in recent years. These electrodes are mainly structurally solid and made from materials such as elemental silver, platinum or gold wires. They are generally used either in miniaturised electrochemical cells (i.e. portable biosensors) or when organic solvents are used as the measurement media [87]. Some of the advantages of pseudo-RE, including simplicity, no liquid gap junction potential, low ohm resistance, no standard RE solvent contamination. However, the inherent potential of pseudo-RE cannot be measured due to the lack of thermodynamic equilibrium, potential shifts occur because many of them are not ideally nonpolarisable, and have limited operating pH or temperature range [89]. Overall, different RE varieties and their respected characteristics could have an impact on and shift the potential of the sample. The RE selection should, therefore take into account the working electrode, the oxidation potential of the sample and the nature of the measurement environment.
This was added from line 287 to 309 in response to the comment. With addition of reference 87-89.
- On page 5, discussing the Faradaic current and electric double-layer current. I suggest that the authors change this to Faradaic and non-Faradaic currents. Non-Faradaic currents can indeed come from double layer capacitance, as they say, but they can also be a result of adsorption/desorption processes.
Changes made, thankyou
- Also in this section, it is mentioned that the scan rate should be selected to provide the optimum balance between Faradaic and EDL currents. The first thought I had is that square-wave voltammetry (SWV) could overcome this – i.e. reduce the effect of non-Faradaic currents. I understand that the review is focussed on cyclic voltammetry, but the use of SWV seems obvious for sensitive detection of analytes, and could be a recommendation. This point also relates to the last sentence in the discussion that says that voltammetric data from SWV are less easy to interpret than standard CV. I actually disagree with this. CV is used to study mechanisms and to extract kinetic and thermodynamic parameters, and is very powerful because of this. SWV is used for electroanalysis of low concentrations once you know what peak you are analyzing. Of course, the choice of CV vs SWV will depend on how sensitive the measurements are. Some more discussion on this would be beneficial.
However, due to its relative complexity, voltammetric data from SWV are less easy to interpret than standard CV, where in most cases, only the net voltammetric peak is analyzed [100]. Consequently, given that CV is a powerful technique for investigating kinetics and mechanisms of the electrochemical process, it is limited because of the lower sensitivity of analyte detection when compared to SWV [101]. An ideal analytical study should therefore begin by performing a CV scan within a large potential window to explore the nature of redox species and determine the potential for oxidation and reduction. This is followed by a highly sensitive SWV scan to analyze the compounds of interest in a smaller, predetermined potential ranges.
Changes and additional discussion added between line 359 to 367 with the addition of reference 101.
- Please also use the conventional abbreviation SWV for square wave voltammetry, not SQV. Correction made, thankyou
- Please remove section 0 and the last part of the discussion section, that are instructions from the journal
Correction made thankyou
Reviewer 2 Report
In this manuscript, the authors exclusively summarized 38 studies that used cyclic voltammetry to determine the change of antioxidant status in humans and animals, including the sample preparation methods, sample processing and storage, experimental details and the techniques to identify the antioxidants corresponding to the voltammetric peaks. Based on the existed studies, some recommendations to the experimental procedures have been provided. However, the 38 studies included in this review used various test conditions as summarized in Table 1 (different organisms, sample type and voltammetry technique details). All the specific cyclic voltammetry technique used to determine an antioxidant is from an individual research. There is no evidence that the unique technique is repeatable or reliable by other independent researches. The authors should address this issue as current cyclic voltammetry technique is insufficient to be an important tool for biological and clinical applications, unless more researches arise.
Minors: a space is need between number and unit, like -70 0C. Same as "n = 2". Please revise them for the entire manuscript.
Author Response
Thanks for the time and kind comments you made. As per your advice, we have revised the paper.
In this manuscript, the authors exclusively summarized 38 studies that used cyclic voltammetry to determine the change of antioxidant status in humans and animals, including the sample preparation methods, sample processing and storage, experimental details and the techniques to identify the antioxidants corresponding to the voltammetric peaks. Based on the existed studies, some recommendations to the experimental procedures have been provided. However, the 38 studies included in this review used various test conditions as summarized in Table 1 (different organisms, sample type and voltammetry technique details). All the specific cyclic voltammetry technique used to determine an antioxidant is from an individual research. There is no evidence that the unique technique is repeatable or reliable by other independent researches. The authors should address this issue as current cyclic voltammetry technique is insufficient to be an important tool for biological and clinical applications, unless more researches arise.
Overall, in this current review, given the multidisciplinary nature of CV research, we focused on the technical aspect of using CV with biological samples. The study will likely assist researchers with a clinical background who have difficulty grasping the concept of electrochemistry and those with electrochemistry background who are not experienced in the handling of biological samples.
The clinical evidence of the use of this technique for in vivo oxidative stress or antioxidant status monitoring is not covered in the current review. CV studies have been used for several medical conditions. These include abetalipoproteinemia [50], ageing [33,54,55], anchored temporomandibular joints [58], cancer [34], cerebral ischemia [48], colitis [42], CuZn-SOD deficiency [40], diabetes [46,60], dietary restriction [45], drug toxicity [66-70,77], duodenal ulcer [56], gastric dilatation [62], glycogen storage [62]. Since the majority of studies are limited in the scale, and the research conditions vary considerably, further research is needed to support the translation of the CV technique into full clinical use.
This was added between 368 to 379 in response to the comment.
- Minors: a space is need between number and unit, like -70 0C. Same as "n = 2". Please revise them for the entire manuscript.
Correction made, thank you.
Reviewer 3 Report
This study presents a review of the measurement of oxidative stress using cyclic voltammetry. I think the manuscript is suitable for publication in this journal after some changes are made.
The comments are here,
- Materials and methods are well mentioned. However, I did not see any results in the result section, rather it also seems like methods. Please eye on that.
- In original papers, there should be the results presented in terms of concentration in biological samples, however, the authors did not mention any key results in this review except section 3.3.
- The authors should include or explain the electrolytes used during measurement.
- Controlling the interference by other products during measurement of the targeted marker is challenging, How this could be managed in the reported literature. Please include in revised manuscript.
- Did other reports compare their values obtained from CV with the values obtained from author techniques? Should explain.
- A minor mistake in line 134.
Author Response
Thanks for the time and kind comments you made. As per your advice, we have revised the paper.
The authors should include or explain the electrolytes use This study presents a review of the measurement of oxidative stress using cyclic voltammetry. I think the manuscript is suitable for publication in this journal after some changes are made.
The comments are here,
- Materials and methods are well mentioned. However, I did not see any results in the result section, rather it also seems like methods. Please eye on that.
This is a systematic review rather than an original paper. The data we extracted from the past papers is the result section. The paper aims to guide future researchers on using the technique for measuring the biological samples.
- In original papers, there should be the results presented in terms of concentration in biological samples, however, the authors did not mention any key results in this review except section 3.3.
This is a systematic review rather than an original paper. The data we extracted from the past papers is the result section. The paper aims to guide future researchers on using the technique for measuring the biological samples.
- The authors should include or explain the electrolytes used during measurement
Blood samples have been measured neat, where the electrolytes required for the measurement are provided intrinsically. If the blood samples were neatly measured, the electrolytes needed for the measurement are supplied intrinsically. Alternatively, some researchers have reported using the Phosphate Buffer Saline (PBS) or KCl to dilute their sample [31,32,34,76]. In the case of PBS, it will supply the measurement with additional electrolytes, including NaCl, KCl, Na2HPO4, KNPO4.
This was added between 152 to 156 in response to the comment.
- Controlling the interference by other products during measurement of the targeted marker is challenging. How this could be managed in the reported literature. Please include in the revised manuscript.
Section 3.3 shows how these papers were trying to figure out what the voltammogram of biological samples represents. See the discussion from line 345- 353 with a minor alteration of the text from 350-353 to address the point.
- Did other reports compare their values obtained from CV with the values obtained from author techniques? Should explain.
This is a systematic review rather than an original paper. The data we extracted from the past papers is the result section. Table 3 has listed the CV potential obtained by these papers and corresponding antioxidants.
- A minor mistake in line 134.
Line 134 revised.